# The Effects of Tricalcium-Silicate-Nanoparticle-Containing Cement: In Vitro and In Vivo Studies

**DOI:** 10.3390/ma16124451

**Published:** 2023-06-18

**Authors:** Naho Ezawa, Yoshihiko Akashi, Kei Nakajima, Katsutoshi Kokubun, Masahiro Furusawa, Kenichi Matsuzaka

**Affiliations:** 1Department of Endodontics, Tokyo Dental College, Tokyo 101-0061, Japan; mfurusawa@tdc.ac.jp; 2Department of Pathology, Tokyo Dental College, Tokyo 101-0061, Japan; akashiyoshihiko@tdc.ac.jp (Y.A.); nakajimakei@tdc.ac.jp (K.N.); kkokubun@tdc.ac.jp (K.K.); matsuzak@tdc.ac.jp (K.M.)

**Keywords:** tricalcium-silicate-nanoparticle-containing cement (Biodentine), mineral trioxide aggregate (MTA), human periodontal ligament fibroblasts (HPLFs)

## Abstract

A tricalcium-silicate-nanoparticle-containing cement (Biodentine) was developed to overcome the disadvantages of existing mineral trioxide aggregate (MTA) dental materials. This study aimed at evaluating the effects of Biodentine on the osteogenic differentiation of human periodontal ligament fibroblasts (HPLFs) in vitro and the healing of furcal perforations created experimentally in rat molars in vivo, in comparison to MTA. The in vitro studies performed the following assays: pH measurement using a pH meter, the release of calcium ions using a calcium assay kit, cell attachment and morphology using SEM, cell proliferation using a coulter counter, marker expression using quantitative reverse transcription polymerase chain reaction (qRT-PCR) and cell mineralized deposit formation using Alizarin Red S (ARS) staining. In the in vivo studies, MTA and Biodentine were used to fill the rat molar perforations. Rat molars were processed at 7, 14 and 28 days for analysis of inflammatory processes using hematoxylin and eosin (HE) staining, immunohistochemical staining of Runx2 and tartrate-resistant acid phosphate (TRAP) staining. The results demonstrate that the nanoparticle size distribution of Biodentine is critical for osteogenic potential at an earlier stage compared to MTA. Further studies are required to elucidate the mechanism of action of Biodentine in osteogenic differentiation.

## 1. Introduction

One of the complications of root canal treatments is accidental perforation, and, in particular, when the pulp chamber floor is perforated, the prognosis is often poor [1]. Our department has compared and examined various perforation site sealants in order to improve the prognosis for perforations at the floor of the pulp chamber [2,3].

When the floor of the pulp chamber is perforated, a calcium-silicate-based cement, such as mineral trioxide aggregate (MTA), is generally used as a perforation sealant [4,5,6]. Previous studies have demonstrated favorable periradicular tissue response to the use of MTA in furcation perforations and induction of the formation of mineralized tissue [7,8,9]. It has been reported that when MTA is co-cultured with human periodontal ligament cells, osteoprotegerin (OPG) and osteocalcin (OCN) mRNAs are expressed [10]. It has also been suggested that MTA promotes the differentiation of human periodontal ligament fibroblasts (HPLFs) into osteoblast/cementoblast-like cells [10]. However, MTA has a long setting time and is also difficult to manipulate and insert [5]. Moreover, there is evidence that bismuth oxide, used as a radiopacifier in dental materials, reduces compression resistance and inhibits cell proliferation [11,12]. 

A tricalcium-silicate-nanoparticle-containing cement (Biodentine) was developed to overcome the disadvantages of existing MTAs, first marketed in 2009 [13,14,15]. Biodentine is technically based on MTA technology. MTA and Biodentine are generally classified as bioceramic cements. Biomaterials are defined as materials that are in direct contact with various tissues in the body (fluid, soft and hard tissues). Bioceramic cements are characterized by biocompatibility and durability [15,16]. The first bioceramic material successfully used in endodontics was MTA, which was developed in the early 1990s based on Portland cement. It was developed as a retrograde filling material and for perforation closure [15,16]. Biodentine has a superior operability, independent of the dentist’s skill, and biocompatibility for a wide range of applications including endodontic repair, and it has attracted increasing attention in recent years as a cement designed as a bioactive dentin substitute, which has mechanical properties similar to dentin [13,17,18]. Biodentine has also been shown to have a faster setting time compared to MTA because the liquid contains calcium chloride and water-soluble polymers that reduce the time required for setting [13,19,20]. Many studies have been conducted in the past comparing the effects of Biodentine on the gene expression levels of runt-related transcription factor 2 (RUNX2) and receptor activator of nuclear factor-kappa B ligand (RANKL) to MTA in dental pulp cells [21,22,23]. However, studies on the effects of Biodentine on HPLFs are not completely clear. Therefore, the purpose of this study was to evaluate the effects of MTA and Biodentine in a rat model of pulp chamber floor perforation and in HPLFs.

## 2. Materials and Methods

### 2.1. In Vitro Studies

#### 2.1.1. Cells

Commercially available primary HPLFs (ScienCell, #2630, San Diego, CA, USA) isolated from human periodontal tissue were used in this study.

#### 2.1.2. Cell Culture Medium 

Fibroblast medium (FM), which is an HPLF growth medium (#2321, ScienCell, San Diego, CA, USA), was supplemented with 2% fetal bovine serum (FBS), fibroblast growth supplement (FGS-acf, #2372) and 1% antibiotic solution (P/S, #0503). The osteoinductive medium was prepared by adding dexamethasone (10^−7^ mol/L), ascorbic acid (2 × 10^−4^ mol/L) and glycerophosphate (10 mmol/L) to the FM.

#### 2.1.3. Cell Culture 

HPLFs were seeded in 100 mm dishes at a density of 5000 cells/cm^2^ (275,000 cells/dish) and were cultured in FM that was changed every 3 days at a temperature of 37 °C with 5% carbon dioxide. HPLFs were detached using trypsin/EDTA and were passaged. HPLFs that had reached confluence after 3–5 passages were used in this study. The day when the culture medium was changed is defined as day 0 (Figure 1).

#### 2.1.4. Production of Material Disks Containing MTA and Biodentine

The following tricalcium-silicate-based cement materials were used to prepare extracts: (i) ProRoot^®^ MTA (Dentsply Tulsa Dental Specialties, Tulsa, OK, USA) containing Portland cement (tricalcium silicate, dicalcium silicate and tricalcium aluminate) 75%, calcium sulfate dihydrate (gypsum) 5% and bismuth oxide 20%, and (ii) Biodentine^®^ (Septodont, Saint-Maur-des-Fossés, France) containing tricalcium silicate 80.1%, calcium carbonate 14.9% and zirconium oxide 5.0%.

MTA and Biodentine were obtained from dental material handling manufacturers. All cements were mixed according to the manufacturer’s guidelines under aseptic conditions and were molded into 7 mm diameter and 1 mm thickness disks. The sample size was estimated based on data from previous studies of similar design [24,25]. All disks were placed in a 100% humidity incubator at 37 °C for 24 h, then were immersed in 70% ethanol and sterilized with UV light for 30 min at room temperature. All disks were placed in the center of each well in 6-well plates.

#### 2.1.5. pH Measurement

A disk of each material in FM (pH 7.2, 2 mL) was placed in an atmospheric environment for 3 days. The pH of the medium was evaluated using a digital pH meter (LAQUA twin pH, HORIBA, Kyoto, Japan). As a control, FM was measured (n = 5).

#### 2.1.6. Calcium Ion Release 

Calcium ion release from the materials was measured in the presence or absence of HPLFs. HPLFs (5.0 × 10⁴) were seeded in 6-well culture plates in a 100% humidity incubator at 37 °C for 3 days. After day 3, the concentration of calcium ions in the medium was measured. Calcium ion release was measured using a calcium assay kit (Metallogenics, Chiba, Japan). Calcium ion release in the medium was rapidly quantified in a microplate reader (96 wells) and the calcium ion concentration was calculated. As a control, FM was measured in the presence or absence of HPLFs (n = 5).

#### 2.1.7. Scanning Electron Microscope (SEM) Observations

Observations of material particle size, disk surface morphology and cell adhesion were performed using SEM (Hitachi S-4800, Hitachi High-Technologies Corp., Tokyo, Japan) (at 1.00 k × magnification, 15 kV). Disks made of each material were placed in 6-well plates, and HPLFs (1.0 × 10⁶) were seeded on those disks and cultured in a 100% humidity incubator at 37 °C for 3, 6, 10 and 14 days. After the incubation period, the material disks along with cells growing on their surfaces were washed twice with PBS and were fixed overnight in Karnovsky’s fixative (2.5% glutaraldehyde, 2% formaldehyde, 0.1 M sodium cacodylate). The disks were then dehydrated in a graded series of ethanol (70–95% *v*/*v*), mounted on a sample stage and coated with gold/palladium prior to SEM observation (n = 5).

#### 2.1.8. Proliferation of HPLFs 

HPLFs (5.0 × 10⁴ cells) were seeded in 6-well plates and then exposed to each type of cement in a 100% humidity incubator at 37 °C. At days 1, 3 and 5, cell proliferation was evaluated using a Coulter Counter (Beckman Coulter, Tokyo, Japan) and the numbers of cells were calculated. As a control, cells were not exposed to any cement (n = 5). The FM was changed every 3 days.

#### 2.1.9. Quantitative Reverse-Transcription Polymerase Chain Reaction (qRT-PCR) 

HPLFs that had been seeded in osteoinductive medium at days 3 and 6 in each group were collected. Total RNAs were extracted using an RNeasy Mini Kit (QIAGEN, Hilden, Germany) and were reverse transcribed into complementary DNAs (cDNAs) using ReverTra Ace qPCR RT Master Mix with gDNA Remover (TOYOBO, Osaka, Japan). qRT-PCR was performed using TaqMan Gene Expression Assays (Applied Biosystems, Waltham, MA, USA). The target genes characterized in this study as osteoblast markers were: RUNX2 mRNA (Hs01047973-m1), OCN mRNA (Hs01587814-g1), RANKL mRNA (Hs00243522-m1) and OPG mRNA (Hs00900358-m1). Glyceraldehyde-3-phosphate dehydrogenase (GAPDH: Hs02786624_g1) was used as an endogenous control. qRT-PCR was performed using a 7500 Fast Real-Time PCR System (Applied Biosystems, Waltham, MA, USA). mRNA expression levels were corrected based on GAPDH mRNA expression levels, and target gene expression levels were subjected to relative quantitative analysis. As a control, osteoinductive medium was measured in the presence of HPLFs (n = 5).

#### 2.1.10. Alizarin Red S (ARS) Staining 

ARS staining was used to assess mineralized deposit formation in cell cultures. The purpose of carrying out ARS staining is to quantitatively evaluate the amount of calcium deposited by the calcium silicate cement. HPLFs (1.0 × 10⁶) were seeded in osteoinductive medium (2 mL) in 6-well culture plates containing material disks (n = 5) for 24 days. The osteoinductive medium was changed every 3 days. After day 24, HPLFs were fixed with 2% paraformaldehyde (Nacalai Tesque, Inc, Kyoto, Japan) for 10 min at room temperature. After fixation, each well was washed twice with purified water and 1 mL ARS staining solution (pH 6.4) was added. After standing at room temperature for 10 min, the ARS staining solution was aspirated from the wells, and the wells were washed 3 times with purified water (1 mL each). The wells were photographed immediately after drying in air (Canon Inc., Tokyo, Japan). Results were analyzed statistically by measuring the stained area within each photograph.

### 2.2. In Vivo Studies

#### 2.2.1. Experimental Design 

The study protocol complied with the Guidelines for the Treatment of Experimental Animals at Tokyo Dental College (Approval Number: 200502). In this study, 36 healthy male Wistar rats, 10 weeks old and each weighing an average of 300 g (Sankyo Lab Service, Tokyo, Japan), were used. The animals were in good health at the beginning of the experiment.

The sample size was estimated based on data from previous studies of similar design [26]. Using an alpha error of 0.05% and 95% power to recognize a significant difference of 1 in the median scores, a minimum of 9 animals per group was considered necessary. Taking into consideration the possible complications that could occur during the study, one more animal was added in each group for each experimental time. Considering three observation times, 36 animals were necessary. Rats were anesthetized using three kinds of mixed anesthesia (1.5 mL/mg, Domitor (ZENOAQ, Fukushima, Japan), Midazolam (SANDOZ, Tokyo, Japan) and Butorphanol (Meiji Seika, Tokyo, Japan)). Perforation of the pulp chamber floor was performed according to the method described by Nakauchi et al. [27]. Briefly, a cavity was prepared on the occlusal surface of the upper first molar of each rat using a 0.5 mm round burr (MANI#1/4, INC, Tochigi, Japan). The pulp chamber floor was then perforated toward the periodontal ligament (PDL) without injuring the alveolar bone. The perforation was confirmed using a stereoscopic microscope(Carl Zeiss, Oberkochen, Germany). The coronal access was performed at low speed with saline solution irrigation. Hemorrhages were arrested using physiological saline irrigation and sterile cotton pellets. All materials were gently placed with a curette until they completely filled the furcation perforation (shown schematically in Figure 2A). After the initial setting time, all coronal access cavities were restored with a light-curing glass ionomer cement (GIC, DMG, Chemisch-Pharmazeutische Fabrik GmbH, Berlin, Germany). At the end of the experimental period, rats were euthanized with an anesthetic overdose. During the experiment, all rats were housed under a 12 h light/12 h dark cycle at a controlled temperature (22 °C), with water and food provided ad libitum. The general health of rats was monitored throughout the experimental period. The groups were divided into three groups of each material at 7, 14 and 28 days (Control, MTA and Biodentine groups). The control group was GIC only (n = 4).

#### 2.2.2. Histological Observations 

At days 7, 14 and 28 after surgery, the maxillary first molar of each rat was excised together with the maxillary bone. The excised maxilla was immersed and fixed in 4% paraformaldehyde at 4 °C for 24 h, decalcified with 10% EDTA (Fuji Film, Tokyo, Japan) at 4 °C for 1 month and then embedded in paraffin. The EDTA was changed 3 times a week. Paraffin blocks were sectioned approximately 5 μm thick sagittally to the tooth. After sectioning, the sections were deparaffinized and stained with hematoxylin and eosin (HE). We observed three areas: the PDL at cement areas, the PDL at root areas and the PDL at alveolar bone areas. The PDL at cement areas shows the PDL at the furcation area attached to the cement. The PDL at root areas shows the PDL at the coronal side of the tooth root. The PDL at alveolar bone areas shows the PDL at furcation area attached to the alveolar bone ridge (Figure 2B).

#### 2.2.3. Immunohistochemical Staining of Runx2, and Ratio of Runx2-Positive Cells

For immunohistochemical observations, the paraffin sections were deparaffinized with xylol, after which they were activated for antigen using a tryptic antigen retrieval kit (Abcam, Cambridge, UK) for 10 min. Each section was immersed in methanol containing 0.3% aqueous hydrogen peroxide at room temperature to block endogenous peroxidase activity, then blocked for 30 min with 10% goat serum at room temperature to reduce non-specific binding. Sections were reacted with the primary antibody overnight at 4 °C. The primary antibody used was Runx2 (sc-390351, 1:50; Santa Cruz Biotechnology, Dallas, TX, USA). The sections were then reacted with MACH 2 Universal HRP Polymer Detection (BRR522G, BIOCARE MEDICAL, Pacheco, CA, USA) as a peroxidase-conjugated secondary antibody for 30 min at room temperature, after which they were stained with 3,3′-diaminobenzidine (DAB), and nuclei were stained with hematoxylin. As a negative control, the primary antibody was replaced with goat nonimmune serum. The PDL at the root area (100 µm × 100 µm), as shown in Figure 2B, was observed with a light microscope(Carl Zeiss, Oberkochen, Germany), and the number of Runx2-positive cells was counted. The ratio of Runx2-positive cells was calculated as Runx2-positive cells/total cells × 100 (%). 

#### 2.2.4. Tartrate-Resistant Acid Phosphate (TRAP)-Positive Staining, and the Number of TRAP-Positive Cells

The histochemical TRAP reaction was used as an osteoclast marker (Fuji Film, Tokyo, Japan). After deparaffinization and washing with water, the TRAP staining solution (0.5 mL, pre-adjusted) was incubated on each section for 30 min at room temperature, after which the sections were washed in distilled water, counterstained with hematoxylin and mounted in aqueous medium. The number of TRAP-positive cells was determined as previously described [28,29]. TRAP-positive cells were counted within the PDL at alveolar bone areas corresponding in length to the perforation area (Figure 2B). In each section, an image of the interradicular alveolar process was captured using a camera attached to a light microscope. Subsequently, the numbers of multinucleated TRAP-positive cells adjacent to the alveolar process surface were counted using a light microscope and were divided by the total length of the bone perforation surface [21].

#### 2.2.5. Statistical Analysis

Quantitative data are expressed as means ± standard deviation (SD) using GraphPad Prism7 (version 7 for Windows, MDF Corp., Tokyo, Japan). Quantitative data were analyzed via relative evaluation using Student’s *t*-test and one-way ANOVA analysis with post hoc Tukey’s multiple comparison test (* *p* < 0.05). 

## 3. Results

### 3.1. In Vitro Studies

#### 3.1.1. pH Measurement

The mean values and standard deviations of pH observed for each type of cement tested are shown in Table 1. The control pH was 8.28 ± 0.01, the MTA pH was 8.88 ± 0.01 and the Biodentine pH was 8.82 ± 0.00. The pH values of the MTA and Biodentine groups were higher than the control group. There was no significant difference between the MTA and Biodentine groups.

#### 3.1.2. Calcium Release 

In the absence of HPLFs, calcium ion levels in the Biodentine group were significantly higher than in the control and MTA groups (Figure 3). In the presence of HPLFs, the calcium ion levels in the MTA and Biodentine groups were significantly higher than in the control group, and there were no significant differences between the MTA and Biodentine groups. Interestingly, calcium ion levels in the presence of HPLFs were significantly reduced in the Biodentine group compared to calcium ion levels in the absence of HPLFs.

#### 3.1.3. SEM Analysis 

MTA and Biodentine particles had uneven shapes (Figure 4A,B). In addition, the MTA and Biodentine disk surfaces had uneven shapes, and the MTA granules had a round shape while the Biodentine granules had a square shape. HPLFs that were attached on each experimental disk at day 3 were similar, with a spindle shape (Figure 4C,D). HPLFs that were attached on each experimental disk at day 6 showed multiple process extensions in the Biodentine group compared to the MTA group (Figure 4E,F). At days 10 and 14, HPLFs were observed to overlap in multiple layers on the disk surface of both types of cement (Figure 4G–J).

#### 3.1.4. Proliferation of HPLFs 

HPLFs exposed to MTA or Biodentine showed proliferation rates that were statistically similar to the control (Figure 5). A significant increase in proliferation was observed in the MTA group compared to the control group at day 5.

#### 3.1.5. qRT-PCR 

The mRNA expression level of RUNX2 was significantly higher in the MTA and Biodentine groups at day 6 compared to the MTA and Biodentine groups at day 3 (Figure 6A). Although over a period of 3 days, there was an increase in the expression of OCN mRNA in the MTA and Biodentine groups, this higher level was statistically significant only at day 6. Furthermore, the mRNA expression level of OCN in the Biodentine group was significantly higher than in the MTA group at day 6 (Figure 6B). The mRNA expression level of RANKL was significantly lower in the MTA group at day 3 compared to the control group. However, the mRNA expression level of RANKL at day 6 was not significantly different among the three groups at day 6 (Figure 6C). The mRNA expression level of OPG was significantly higher in the Biodentine group at day 6 compared to the Biodentine group at day 3. The mRNA expression level of OPG at days 3 and 6 was not significantly different among the three groups (Figure 6D).

#### 3.1.6. Mineralization and ARS Staining Analysis 

After 24 days of HPLF culture on the different materials in osteogenic medium, the MTA and Biodentine groups had increased amounts of calcium deposits (Figure 7A). In particular, the Biodentine group produced significantly more calcified deposits than the control group and the MTA group (Figure 7B).

#### 3.1.7. Histological Observations

At day 7, inflammatory cell infiltrations and multinucleated giant cells were observed in the control group (Figure 8A). Inflammatory cells and a few osteoblasts were observed in the MTA and Biodentine groups (Figure 8B,C). At day 14, inflammatory cells, multinucleated giant cells and epithelial layers were observed in the control group (Figure 9A). A small number of inflammatory cells and a line of osteoblasts were observed in the MTA and Biodentine groups (Figure 9B,C). At day 28, a small number of inflammatory cells, osteoblasts and epithelial layers were observed in the control group (Figure 10A). A few inflammatory cells and a line of osteoblasts were observed in the MTA and Biodentine groups (Figure 10B,C). No epithelial layer was observed in the MTA and Biodentine groups. 

#### 3.1.8. Immunohistochemical Observations of Runx2, and the Ratio of Runx2-Positive Cells 

At day 7, Runx2-positive cells were observed in the Biodentine group but not in the control or MTA groups (Figure 11A–C). At days 14 and 28, a small number of Runx2-positive cells were observed in the control group, and numerous Runx2-positive cells were observed in the MTA and Biodentine groups (Figure 11D–I). Quantitative analysis revealed that the number of Runx2-positive cells was highest in the Biodentine group, and significant differences were observed between the MTA and Biodentine groups at day 14 (Figure 12J). Furthermore, significant differences were observed between the control and Biodentine groups at all time points. 

#### 3.1.9. Observations of TRAP Staining and the Number of TRAP-Positive Cells 

At day 7, numerous TRAP-positive cells were observed in the control group, and a small number of TRAP-positive cells were observed in the MTA and Biodentine groups (Figure 12A–C). At day 28, an apparent reduction in the number of TRAP-positive cells was seen in the control group (Figure 12D), while a small number of TRAP-positive cells were observed in the MTA and Biodentine groups (Figure 12E,F). Quantitative analysis revealed that the highest number of TRAP-positive cells was seen in the control group, and no significant differences were observed between the MTA and Biodentine groups at any time point (Figure 12G). At day 28, the number of TRAP-positive cells decreased significantly in the control group compared to day 7. 

## 4. Discussion

Biodentine has a superior operability and biocompatibility compared with MTA and has been reported to be the best material for clinical applications, such as pulp capping, perforation repair, apexification and reverse root canal filling [13]. Biodentine also has the advantage of a very short setting time compared to MTA. The setting time of the material is 12–13 min, which is significantly less than the MTA (2.2 h) [16,19]. Furthermore, it has been reported that using Biodentine as a reverse root canal filling material significantly increases calcium and silicic acid uptake into dentin compared to MTA [28]. ZrO_2_ in Biodentine has been reported to improve biocompatibility [30]. However, most research on Biodentine has been conducted in an in vitro environment, and there are very few in vivo studies. 

In our in vitro study, HPLFs were seeded around the disks, not just on the disks, because in clinical treatments, the effects act on areas that are not in direct contact with the material. In the in vivo study, the material affected HPLFs through the interstitial fluid, so we observed and evaluated areas in contact with or not in contact with the material. 

The calcium ion concentration in the medium showed the highest value in the Biodentine group. Similarly, ARS staining revealed that the Biodentine group produced significantly more calcified deposits than the MTA group. Previous studies investigating cell proliferation and calcification have reported that extracellular alkaline conditions promote the proliferation and calcification of human cementoblasts in vitro [31]. It has been suggested that Biodentine should be kneaded with calcium chloride, which increases the proportion of calcium compared to MTA and results in an increase in calcium ion concentration [32]. Calcium silicate has been reported to cause a highly alkaline environment and to increase the influx of calcium ions into the extracellular environment [33]. It has also been reported that the hydration of Biodentine produces more calcium than MTA, speeding up the rate of calcification and new bone growth [28]. Calcium chloride promotes hydration reactions and calcium phosphate deposition while maintaining a high pH [34]. These results indicate that the high pH environment and Ca leaching brought about by Biodentine affect calcification [25]. These results are also consistent with previous studies showing that calcium-silicate-based cements induce calcification [24,35,36]. On the other hand, MTA and Biodentine have been reported to have an initial cytotoxicity due to the highly alkaline environment and to cause an initial inhibitory effect on cell growth [17]. In this study, cytotoxicity was evaluated by characterizing the cell proliferation rate. All groups had higher viability rates from days 1 to 5, which suggests that MTA and Biodentine are not cytotoxic. Furthermore, our in vivo experiments showed that the PDL of the MTA and Biodentine groups showed a decrease in inflammatory cells with an increase in osteoblasts.

In addition to the HPLF proliferation rate assay, SEM images were acquired to evaluate the physical appearance of HPLFs growing in direct contact on the disks. Biodentine powder components were previously analyzed using a particle size analyzer. It has been reported that nano-sized particles were detected from a study using a particle size analyzer [14]. Incorporating nanoparticles into the powder component may increase the surface area and ionization, which is considered to reduce the setting time [14]. It has also been reported that calcium plays an important role in fibroblast adhesion and that a higher percentage of calcium may result in better cell adhesion [35,36]. The results of this study show that the cell morphology of HPLFs that adhered to Biodentine disks had a typical spindle-shaped morphology, with more spreading cell projections and cell body extensions compared to the MTA group. We speculate that this result is due to the fact that Biodentine contains nanoparticles, which increases the surface area of the disks compared to MTA, allowing cells to stretch more easily, and since Biodentine has a higher percentage of calcium, it allows for better cell adhesion.

The markers used in this study were selected to characterize osteogenic differentiation. RUNX2 is a transcription factor that is involved in regulating osteoblast differentiation and skeletal morphogenesis [37]. RUNX2 is known to be an early marker of osteoblast differentiation [38]. RUNX2 mRNA levels are upregulated in the early stages and are downregulated in the terminal stages [39]. OCN, a non-collagenous protein, also regulates hard tissue calcification and calcium ion homeostasis and is considered a final differentiation marker of bone regeneration [37,39]. RUNX2 mRNA levels were upregulated in the Biodentine and MTA groups from days 3 to 6. Furthermore, the OCN mRNA level at day 6 was significantly higher in the Biodentine group than in the control and MTA groups. These results suggest that Biodentine has a high calcification capacity. Further studies are needed to characterize these effects at other time points. Although there have been many reports on OCN expression in the PDL in response to Biodentine, there are very few studies on Runx2 expression in the PDL in response to Biodentine, so we decided to focus on Runx2 in our in vivo experiments in this study. In vivo immunohistochemical studies showed Runx2-positive cells in the PDL adjacent to the perforation. Cells in the PDL have been reported to differentiate into osteoblasts [40]. Biodentine has a higher amount of calcium content than MTA, and the nanoparticles it contains may have influenced the expression of Runx2 in the PDL in contact with the material because the particles are fine and easily penetrate the perforations in the perforated floor of the pulp chamber. Further, Runx2 is upregulated in the early stages, which is consistent with the results of our in vitro study. In addition, Biodentine may stimulate the expression of cell differentiation factors, including Runx2, to promote osteoblast differentiation and, consequently, bone formation.

RANKL was initially identified as a plasma-membrane-bound ligand that stimulates osteoclast differentiation and bone resorption. RANKL binds to RANK receptors on the surfaces of monocytes and macrophages, inducing their differentiation into mature multinucleated osteoclasts with bone resorption activity [41,42]. The action of RANKL can be inhibited by a natural inhibitor, OPG, which is expressed primarily on some epithelial cells, vascular endothelial cells and lymphoid cells [43]. OPG blocks RANKL/RANK interactions, thereby inhibiting osteoclast activation and bone resorption, and a decrease in the RANKL/OPG ratio was detected in a previous study [44], which is consistent with the results of this study. The results of TRAP staining, used as an osteoclast marker, also showed a decrease in TRAP-positive cells in the Biodentine group from days 7 to 28. Biodentine and MTA have been reported to be involved in the regulation of inflammatory cells and in the promotion of fibroblast and osteoblast differentiation [26]. A decrease in TRAP-positive cells is responsible for decreased bone resorption. These results suggest that MTA and Biodentine induce osteoblast differentiation and activation and demonstrate osteogenic potential.

In our in vivo experiments, inflammatory cells were observed in the PDL at the cement area in all groups at day 7, but inflammatory cells decreased by day 28. In addition, a line of osteoblasts was observed on the alveolar bone surface in the MTA and Biodentine groups from day 14, suggesting that MTA and Biodentine may have promoted osteoblast differentiation. Moreover, an epithelial layer was observed in the control group at days 14 and 28. The perforation of rat molars may stimulate the proliferation of the junctional epithelium [26]. Previous in vivo studies have reported epithelial invasion in MTA and Biodentine [26]. Nakauchi et al. reported that Malassez’s epithelial rest cells normally present in the PDL of the pulp chamber floor may be involved in epithelialization in rats [27]. Considering the results of this study, we believe that the sufficient thickness of the temporary sealing material in this study provided a sufficiently high alkaline environment and prevented the proliferation of epithelial cells in the MTA and Biodentine groups. In summary, this study lays the groundwork for future investigations of Biodentine as a perforation repair material in experimental models. Further clinical trials should be performed to better elucidate the effects of Biodentine in conventional endodontic treatments.

### Limitations of the Study

The present study has certain limitations. For example, for ethical reasons, we were not able to perform an in vivo study using the human periodontal ligament. Therefore, we used the periodontal ligament cells from a rat instead, since it was necessary to conduct both in vitro and in vivo studies to verify the clinical efficacy of Biodentine.

In addition, there was difficulty in performing perforation in a uniform manner, since it is highly dependent on the skill and experience of the operator.

## 5. Conclusions

In our in vitro experiments, the calcium ion concentration in the medium showed the highest value in the Biodentine group. Similarly, ARS staining revealed that the Biodentine group produced significantly more calcified deposits than the MTA group. RUNX2 mRNA levels were upregulated in the Biodentine and MTA groups from days 3 to 6. Furthermore, the OCN mRNA level at day 6 was significantly higher in the Biodentine group than in the control and MTA groups. Our in vivo experiments showed that Biodentine has a higher calcium content than MTA, and the nanoparticles it contains may have influenced the expression of Runx2 in the PDL in contact with the material because the particles are fine and easily penetrate the perforations in the perforated floor of the pulp chamber. Also, Runx2 is upregulated in the early stages, which is consistent with the results of our in vitro study. The results demonstrate that the nanoparticle size distribution of Biodentine is critical for osteogenic potential at an earlier stage compared to MTA. However, further studies are required to elucidate the specific mechanism of action of Biodentine in osteogenic differentiation.

## Figures and Tables

**Figure 1 materials-16-04451-f001:**
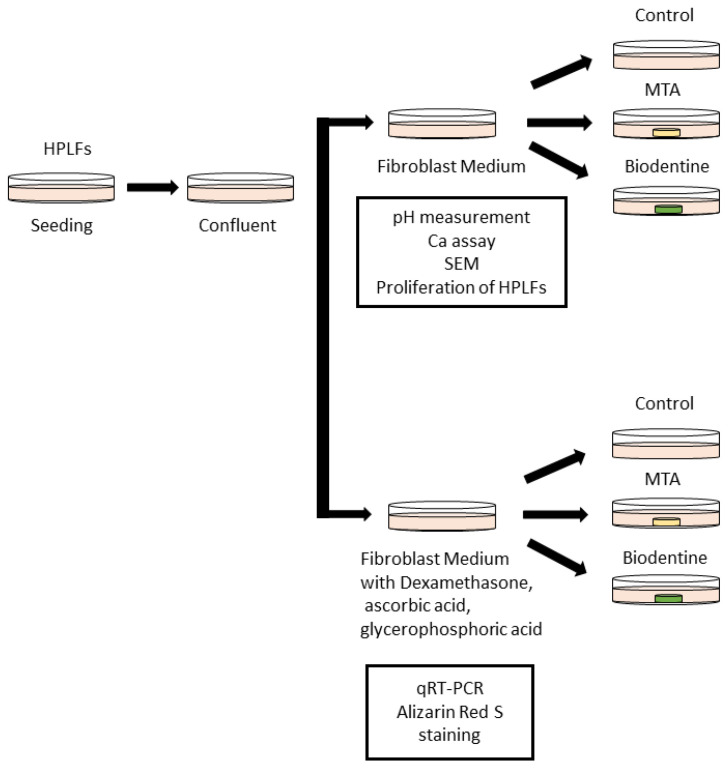
Schematic of the culture method of HPLFs used for the in vitro study. HPLFs were divided into two groups; one group was treated without Dexamethasone, ascorbic acid or glycerophosphoric acid, while the other group was treated with Dexamethasone, ascorbic acid and glycerophosphoric acid.

**Figure 2 materials-16-04451-f002:**
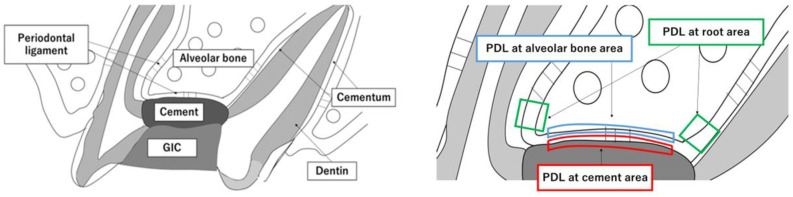
Experimental design of the in vivo study. (**A**) The pulp chamber floor was perforated toward the PDL without injuring the alveolar bone using a 0.5 mm round burr. The perforations were filled with one of the cements. After the initial setting time, all coronal access cavities were restored with GIC. (**B**) Three observation areas. PDL at the cement areas; PDL at the furcation area attached to the cement; PDL at the root areas; PDL at the coronal side of the tooth root (100 µm × 100 µm). PDL at the alveolar bone area; PDL at furcation area attached to the alveolar bone ridge.

**Figure 3 materials-16-04451-f003:**
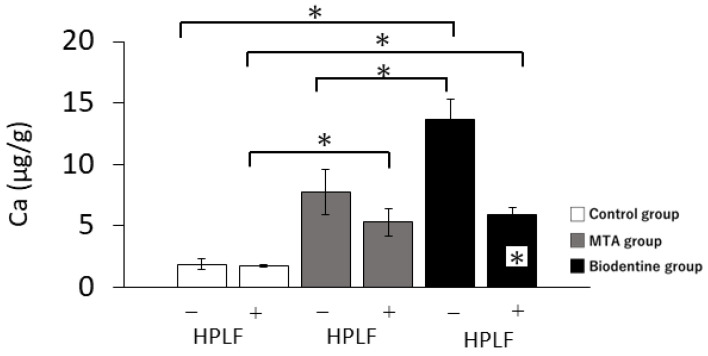
Mean values of calcium ion release from the control, MTA and Biodentine groups over 3 days. The + at the bottom of the graph indicates the presence of HPLFs and the −—indicates the absence of HPLFs. Asterisks in the bars indicate significant differences between the materials (* *p* < 0.05).

**Figure 4 materials-16-04451-f004:**
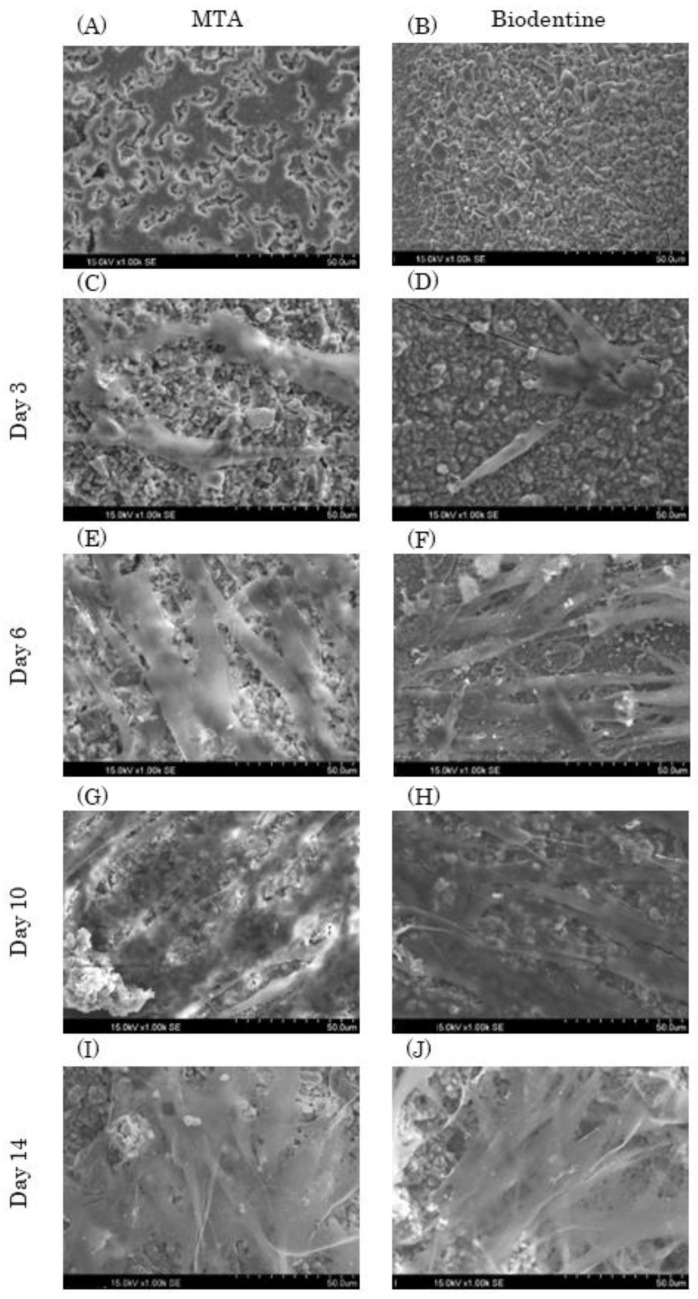
Representative SEM images of MTA and Biodentine with HPLFs (1.00 k × magnification). HPLFs were seeded directly on top of the solid material disks. (**A**,**B**) Each experimental disk without HPLFs. The left column shows images of MTA disks with HPLFs seeded at days 3 (**C**), 6 (**E**), 10 (**G**) and 14 (**I**), and the right column shows images of Biodentine disks with HPLFs seeded at days 3 (**D**), 6 (**F**), 10 (**H**) and 14 (**J**).

**Figure 5 materials-16-04451-f005:**
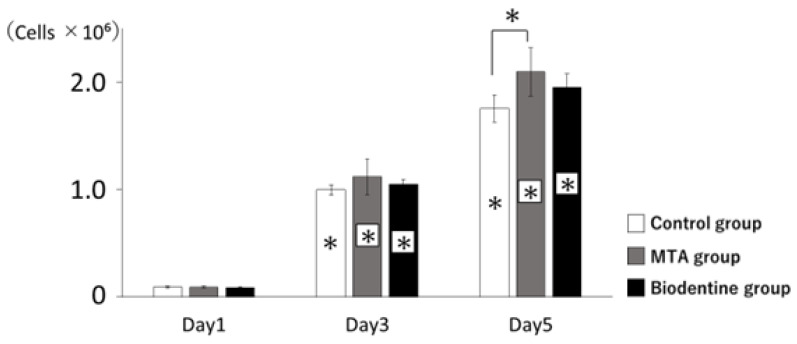
Proliferation of HPLFs. The number of HPLFs at days 3 and 5 showed significantly higher viability rates than at day 1. All groups had higher viability rates from day 1 to day 5 (* *p* < 0.05).

**Figure 6 materials-16-04451-f006:**
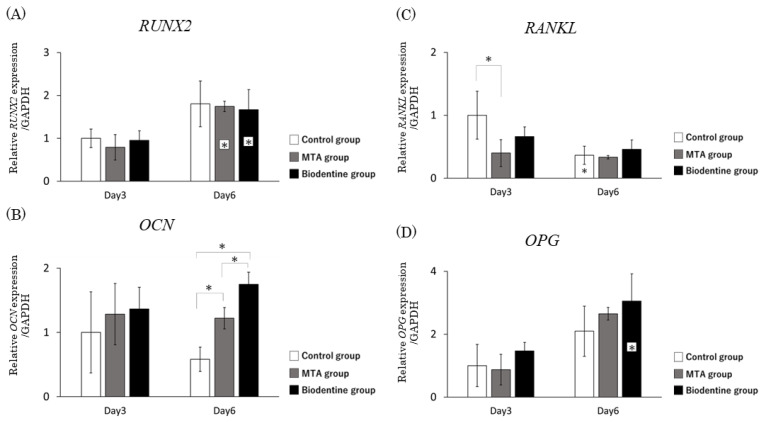
qRT-PCR analysis showed that the mRNA expression levels of HPLFs in vitro were as follows. Asterisks in the bars indicate significant differences between the same materials compared to day 3. (**A**) RUNX2 mRNA expression levels. (**B**) OCN mRNA expression levels. (**C**) RANKL mRNA expression levels. (**D**) OPG mRNA expression levels. (* *p* < 0.05).

**Figure 7 materials-16-04451-f007:**
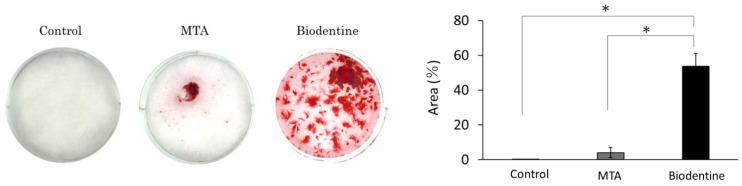
ARS staining assay of MTA and Biodentine media at day 24. (**A**) Representative images of ARS staining assay. (**B**) ARS activity of the MTA and Biodentine media (* *p* < 0.05).

**Figure 8 materials-16-04451-f008:**
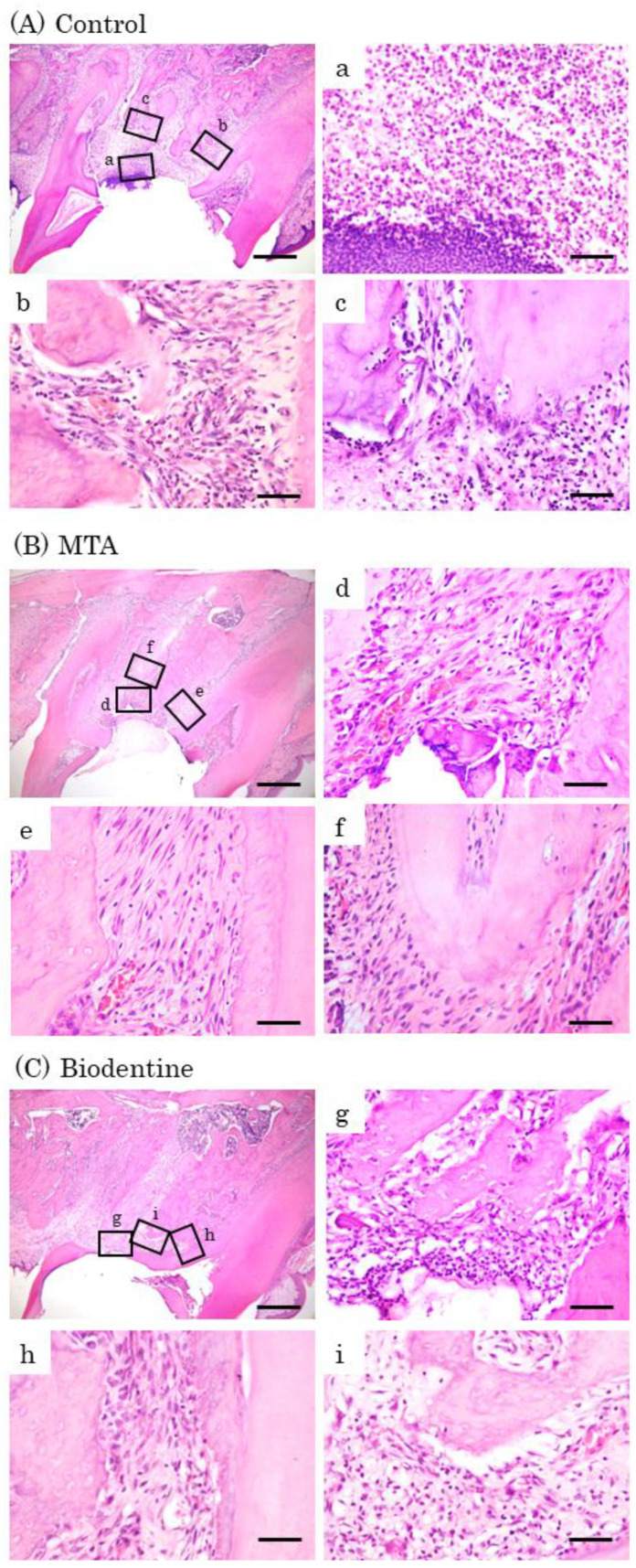
Representative features in the in vivo study using HE staining after surgery at day 7. Light micrographs of sagittal sections of maxilla showing the first molars of the control (**A**), the MTA (**B**) and the Biodentine (**C**) groups. (**A**) Numerous inflammatory cells (**a**–**c**), and multinucleated giant cells were observed (**c**). (**B**,**C**) Inflammatory cells (**d**,**f**,**g**,**i**), and a small number of osteoblasts were observed (**e**,**h**). Scale bars: 500 μm; (**a**–**i**): 50 μm.

**Figure 9 materials-16-04451-f009:**
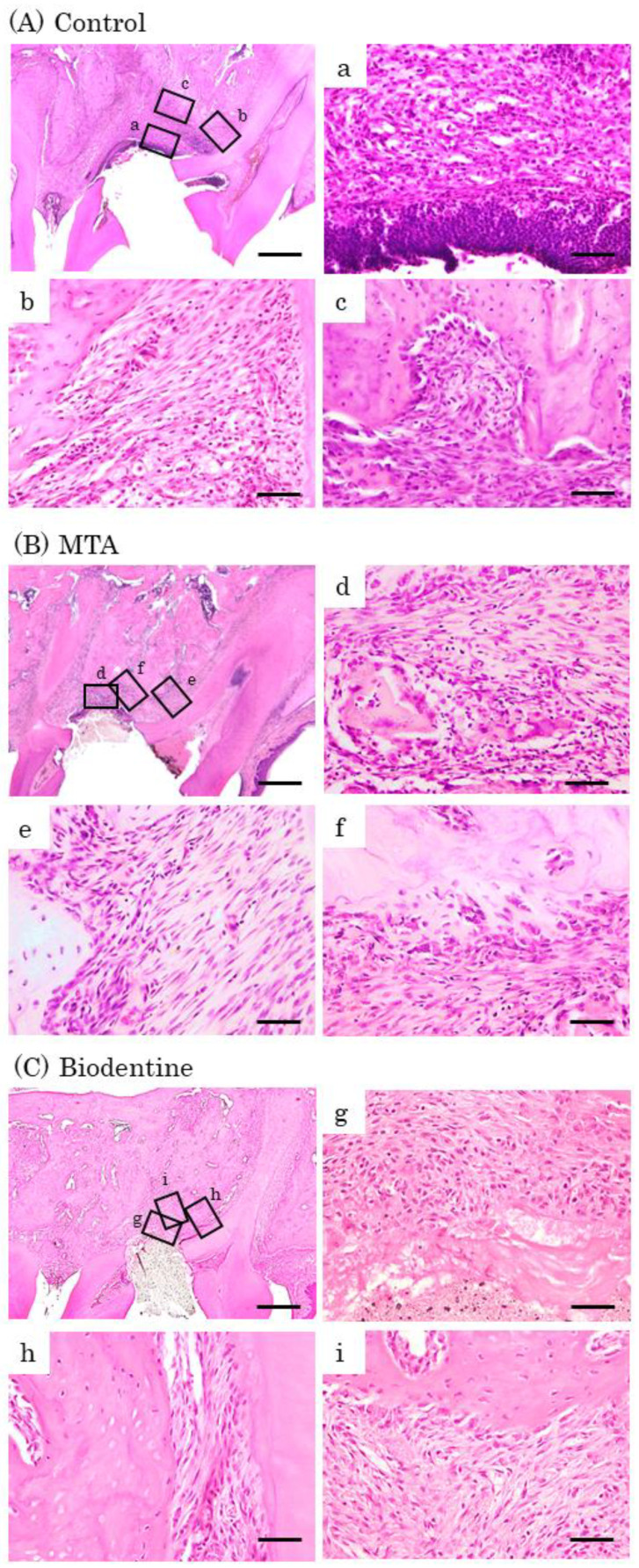
Representative features in the in vivo study using HE staining after surgery at day 14. Light micrographs of sagittal sections of maxilla showing the first molars of the control (**A**), the MTA (**B**) and the Biodentine (**C**) groups. (**A**) An epithelial layer, numerous inflammatory cells (**a**–**c**) and multinucleated giant cells were observed (**c**). (**B**,**C**) A small number of inflammatory cells (**d**,**f**,**g**,**i**), and a line of osteoblasts were observed (**e**,**f**,**h**,**i**). Scale bars: 500 μm; (**a**–**i**): 50 μm.

**Figure 10 materials-16-04451-f010:**
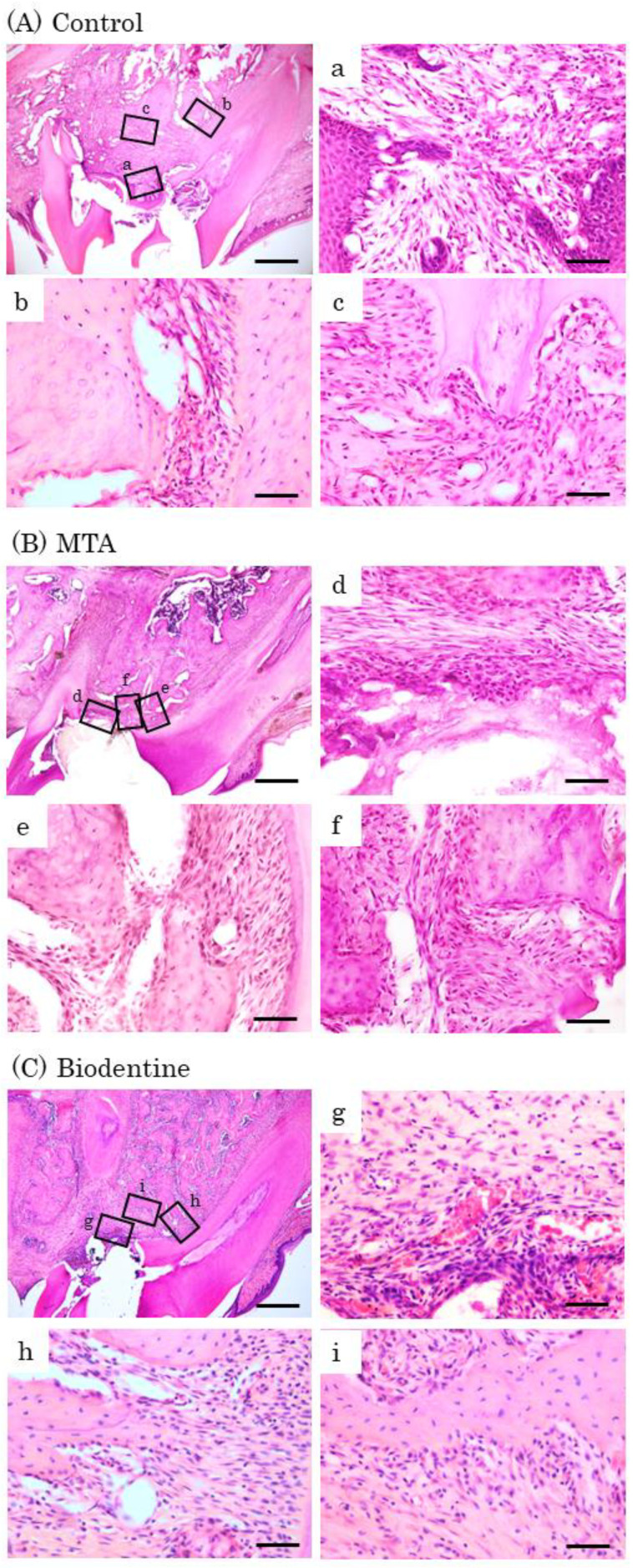
Representative features in the in vivo study using HE staining after surgery at day 28. Light micrographs of sagittal sections of maxilla showing the first molars of the control (**A**), the MTA (**B**) and the Biodentine (**C**) groups. (**A**) An epithelial layer, a small number of inflammatory cells (**a**–**c**) and osteoblasts (**b**,**c**). (**B**,**C**) A few inflammatory cells (**d**,**g**), and a line of osteoblasts were observed (**e**,**f**,**h**,**i**). Scale bars: 500 μm; (**a**–**i**): 50 μm.

**Figure 11 materials-16-04451-f011:**
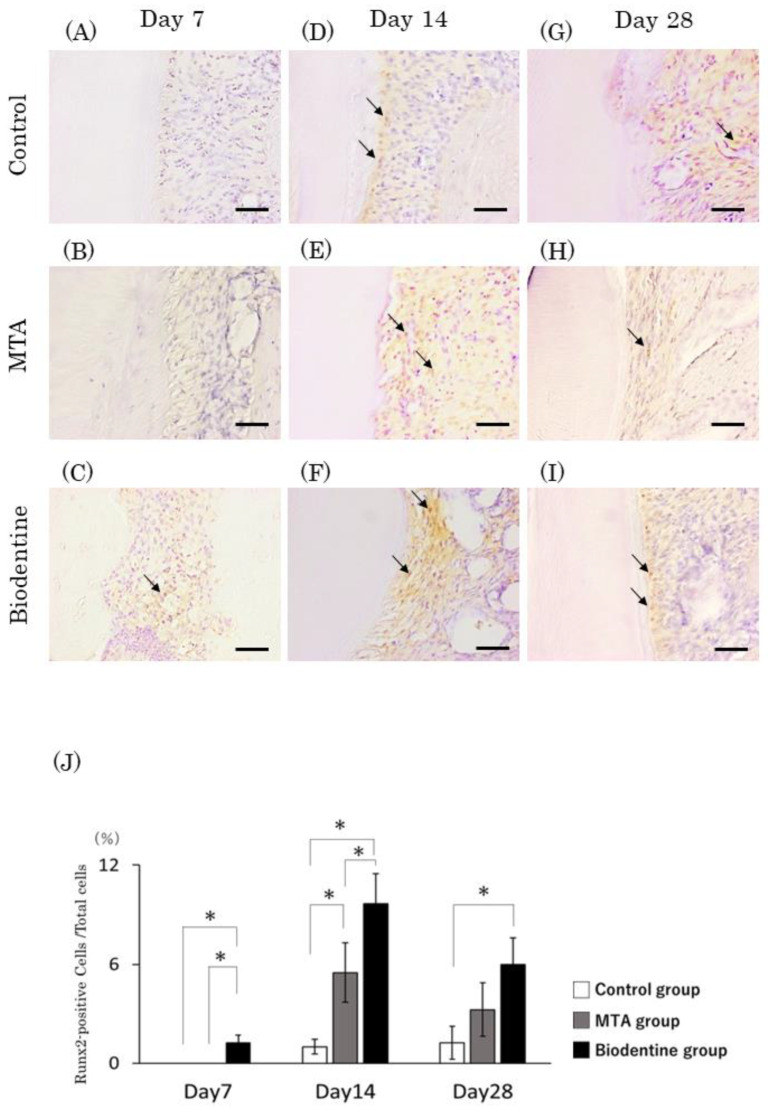
Immunohistochemical staining of Runx2. Light micrographs of sagittal sections of the PDL at root areas of the control (**A**,**D**,**G**), the MTA (**B**,**E**,**H**) and the Biodentine (**C**,**F**,**I**) groups at day 7 (**A**–**C**), at day 14 (**D**–**F**) and at day 28 (**G**–**I**). (**J**) Graph showing the number of Runx2-positive cells in the PDL at root areas from the control, MTA and Biodentine groups at days 7, 14 and 28. (* *p* < 0.05) Arrows; Runx2-positive cells. Scale bars: 50 μm (Arrows; Runx2-positive cells. Scale bars: 50 μm).

**Figure 12 materials-16-04451-f012:**
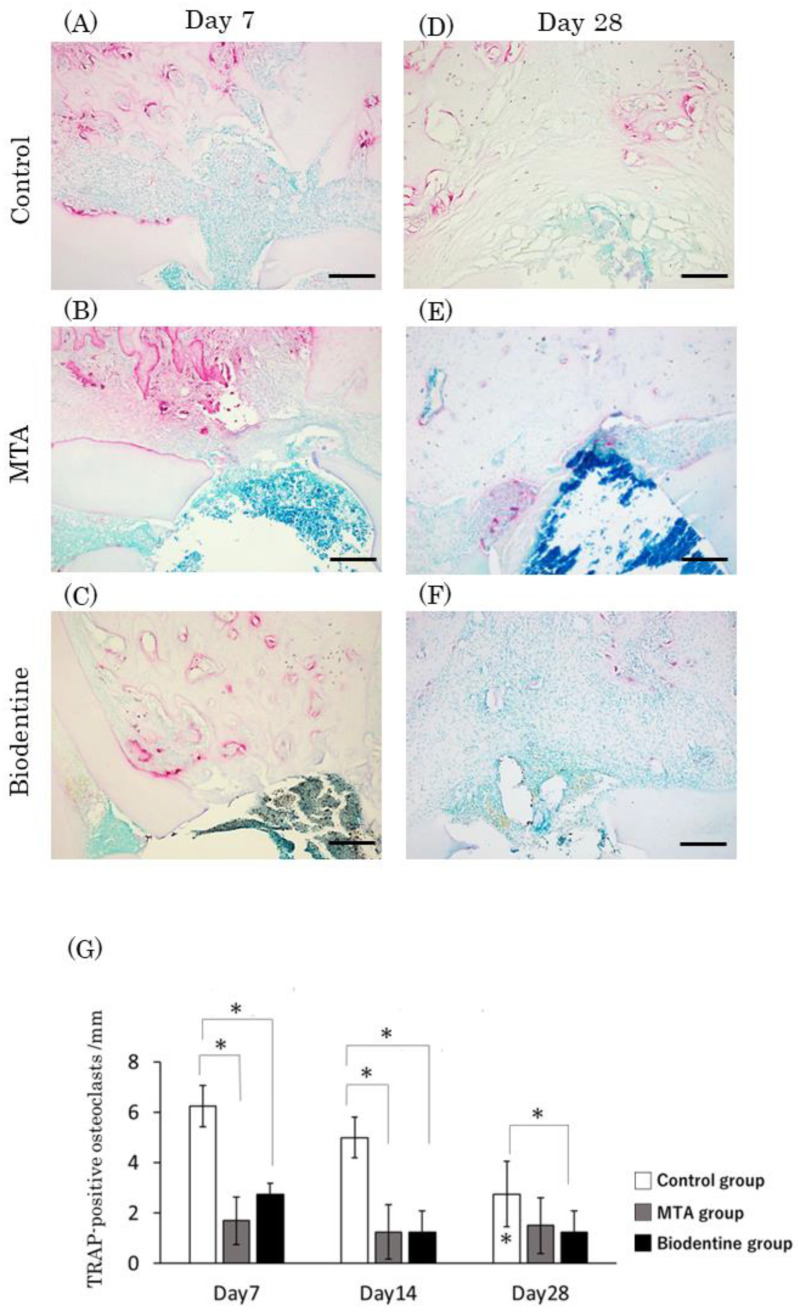
Light micrographs of sagittal sections of the PDL at alveolar bone areas of the control (**A**,**D**), the MTA (**B**,**E**) and the Biodentine (**C**,**F**) groups at day 7 (**A**–**C**) and at day 28 (**D**–**F**). (**G**) Graph showing the number of TRAP-positive cells in the PDL at alveolar bone areas from the control, MTA and Biodentine groups at days 7, 14 and 28. Asterisks in the bars indicate significant differences between the same materials compared to day 7. (* *p* < 0.05) Scale bars: 200 μm.

**Table 1 materials-16-04451-t001:** Mean pH values for control, MTA and Biodentine over 3 days.

pH (Day 3)	
Group	Mean ± SD
Control	8.28 ± 0.01
MTA	8.88 ± 0.01 *
Biodentine	8.82 ± 0.00 *

(* *p* < 0.0001).

## Data Availability

Not applicable.

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
