# Peer review of "The Effects of Tricalcium-Silicate-Nanoparticle-Containing Cement: In Vitro and In Vivo Studies"

_materials, 2023, doi:10.3390/ma16124451_

Round 1
Reviewer 1 Report
The study deals with an interesting topic, however, the presentation is confusing to the readers.
Firstly and most importantly, as the in vitro and in vivo part is not in connected in any ways, they should be presented in 2 different and distinctive manuscripts. This way it is confusing.
The introduction should be lengthened more data on MTA and on furcation defects.
It is not clear: was the original Biodentine used or amodified version??? If Biodentine was used please state this and do not call it NPC.
The manuscript must be brutally restructured base upon these recommendations.
English grammar and spelling checking required.
Reviewer 2 Report
This study is interesting and has scientific value. However, there are a few issues that should be addressed.
— Revise the conclusion "The results demonstrate that the nanoparticle size distribution of NPC is critical for osteogenic potential at an earlier stage compared to MTA". Or add more experiments to demonstrate the nanoparticle size distribution of NPC and its effect on osteogenic potential.
— Add a table to summarize the ingredients of NPC, MTA, and other popular bioceramic cements such as Endosequence BC putty.
— Add more information in the introduction part regarding other popular bioceramic cements.
— Add more new references and discuss NPC, MTA, and other popular bioceramic cements.
— List the limitations of this study in the discussion part.
Minor editing of English language required
Reviewer 3 Report
Dear Authors! Thank you for interesting investigation concerning the in vitro and in vivo study of tricalcium silicate nanoparticle-containing cement.
Several remarks should be made:
The following tables and figures captions contain additional unnecessary information concerning observations and should be rewritten with the removal of the results of observations that are repeated in the text almost verbatim: Table 1, Figure 3, Figure 5, Figure 6, Figure 7(B), Figure 11, Figure 12.
For Figure 8-10, differ-coloured frames for each cells type preferrable for better presentation.
Why were the only male (not female) Wistar rats participating in the experiment?
Line 417: “NPC also has the advantage of a very short setting time compared to MTA [14].” – setting time (minutes) should be noted.
Typos:
Line 82: extra tab
Line 325-329 and 331-335 contains the same text (chapter 3.6).
Line 515: “In our in vivo experiments” should be “in vitro”
Round 2
Reviewer 1 Report
the corrections are fine.
Reviewer 2 Report
No comments.
No comments.